**www.cambridge.org/ext**

**Cite this article:** ÓhÉigeartaigh S (2025). Extinction of the human species: What could cause it and how likely is it to occur? *Cambridge Prisms: Extinction*, **3**, e4, 1–8 https://doi.org/10.1017/ext.2025.4

anthropocene; anthropogenic impacts; climate change; existential risk; societal collapse

**Corresponding author:**
Sean ÓhÉigeartaigh;
Email: so348@cam.ac.uk

# Extinction of the human species: What could cause it and how likely is it to occur?

Sean ÓhÉigeartaigh 🔟

University of Cambridge, Cambridge, UK

## Abstract

The possibility of human extinction has received growing academic attention over the last several decades. Research has analysed possible pathways to human extinction, as well as ethical considerations relating to human survival. Potential causes of human extinction can be loosely grouped into exogenous threats such as an asteroid impact and anthropogenic threats such as war or a catastrophic physics accident. In all cases, an outcome as extreme as human extinction would require events or developments that either have been of very low probability historically or are entirely unprecedented. This introduces deep uncertainty and methodological challenges to the study of the topic. This review provides an overview of potential human extinction causes considered plausible in the current academic literature, experts' judgements of likelihood where available and a synthesis of ethical and social debates relating to the study of human extinction.

## Impact statement

Human extinction may seem unlikely, but cannot be ruled out as a possibility. We now know that there have been cataclysmic events in the past that might have wiped out humans were those events to occur now. Novel threats to human civilisation are now posed by our own activities, both in the form of climate change and environmental degradation and in the form of powerful weapons such as nuclear weapons and emerging technologies such as artificial intelligence (AI) and engineered biology. These risks are the subject of a growing body of research. By reviewing this scholarship, and the evidence for claims that have been made regarding various risks, this review aims to inform the reader of the current state of knowledge in the field. The review aims to be a resource for both academic readers and decision-makers on the state of knowledge regarding a variety of risks – how likely they are and how robust the evidence is. This synthesis will aid experts and decision-makers in assessing how much weight to give concerns regarding human extinction and how to prioritise efforts to reduce or mitigate these risks.

## Introduction

The last half-century has seen growing scientific awareness of natural phenomena such as meteor impacts linked to cataclysmic impacts on Earth, some resulting in mass species extinctions. Scientific evidence has also accumulated around the potentially disastrous consequences of climate change, unsustainable resource use, and the development of powerful technologies, leading scholars to raise concerns that *homo sapiens* may also face imminent extinction threats (Rees, 2003; Bostrom, 2013; Ord, 2020).

As a species, humans have many advantages in the face of such threats. Any catastrophe would need to eliminate virtually all humans across the wide range of environments we inhabit on Earth, leaving remaining humans unable to sustain themselves or reproduce (Avin et al., 2018). Our ability to learn about and respond to the threat would also need to be insufficient. However, there remain potential catastrophes sufficiently large that humans would presently be unable to avoid or defend against them. Developments in science and technology in the last century have made possible the creation of powerful weapons such as nuclear bombs and the tools to coordinate around their wide-scale use. It is likely that further technological breakthroughs will make even more devastating weapons possible and allow them to be used more effectively. Furthermore, it is possible that developments in science and technology may pose risks of accidental disaster or have unintended consequences, such as accelerating humanity's degradation of its environment to the point of uninhabitability (Rees, 2003).

Scholarship on human extinction risk frequently involves discussion of two other closely related terms. *Existential risks* are defined as threatening the premature extinction of Earth-originating intelligent life or the permanent and drastic destruction of its potential for desirable

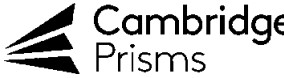



future development (Bostrom, 2013).[1] This category includes extinction risks but also irrecoverable collapse short of full extinction, as well as some more esoteric dystopian scenarios. For the purpose of this review, I will refer to existential risk scenarios that relate to extinction risk, and so the term may be used interchangeably. *Global catastrophic risks* lack a single crisp definition, but generally refer to events or outcomes that could inflict serious damage to human well-being at a global scale, potentially threatening human civilisation[2] (Bostrom and Cirkovic, 2011). Some but not all global catastrophes would lead to human extinction.

The last two decades have seen a growth of scholarship on this class of risks[3] including a number of recent books: *The Precipice* (Ord, 2020) and edited volumes *The Era of Global Risk* (Beard et al., 2023) and *An Anthology of Global Risk* (Beard and Hobson, 2024). *The Precipice* (Ord, 2020) provides analysis of a range of risks to human extinction and the author's assessment of their comparative likelihood. Drawing on the ethical framework of *longtermism*, it argues for the importance of mitigating these risks by emphasising the loss of moral value that would result from the extinguishing of humanity's long-term future. Edited volumes *The Era of Global Risk* (Beard et al., 2023) and *An Anthology of Global Risk* (Beard and Hobson, 2024) encompass a wider range of methodological, ethical, critical, and science policy perspectives on the study of individual risks and global catastrophic risk as a field. Other recent literature provides in-depth historical (Moynihan, 2020; Torres, 2023a,b), philosophical (Torres, 2023a,b; Ware, 2024) and political (Ware, 2024) analyses of the concept of human extinction.

In contrast this review, to the extent possible, limits itself to an up-to-date analysis of the simpler question of "what could kill everyone, and how likely is it to happen?" I will provide a brief synthesis of the current state of knowledge on the most discussed exogenous and endogenous risks, and where available, experts' likelihood estimates. Section 5 provides a brief overview of the complex social, ethical and political debates surrounding the topic, but a detailed treatment is outside the scope of the review.

## Exogenous risks: Broadly unrelated to human activity

Five mass extinctions involving the loss of over 75% of species have occurred in the last 500 million years (Barnosky et al., 2011). In three – the end-Permian, end-Triassic and end-Cretaceous extinctions – flood basalt events are believed to have played a significant role (Bond and Grasby, 2017). These huge lava flows, lasting hundreds of thousands of years (Clapham and Renne, 2019), might conceivably pose extinction risk. However, they are estimated to occur once every 20 to 30 million years (although not at regular intervals) (Courtillot and Renne, 2003). The Toba supervolcanic eruption 74,000 years ago was at one point thought to have come close to wiping out humans, although more recent evidence suggests this is unlikely (Yost et al., 2018). Rougier et al. (2018) suggest a frequency of 60,000–6 million years for Toba-level events;

however, volcanologists remain sceptical that a single volcanic eruption could cause human extinction (Cox, 2017).

Although the Cretaceous-Paleogene extinction event coincides with the Deccan flood basalt eruption (Keller, 2012), the leading candidate for its cause is a different exogenous threat: the ten-kilometre-diameter asteroid that created the Chicxulub crater 66 million years ago (Chiarenza et al., 2020). It is thought an impact of similar magnitude would threaten human extinction (Ord, 2020) although some scholars believe some human populations would survive (Salotti, 2022). By identifying the trajectories of 95% of asteroids >1 km in diameter, asteroid-monitoring efforts have established bounds on the likelihood of a > 1-km asteroid impact in the coming century, estimated at 1 in 120,000 (Ord, 2020).[4] Baum (2018) notes substantial uncertainty with regard to the potential for an asteroid impact to cause civilisational collapse and proposes that 100 m in diameter should be used as a threshold when considering asteroid impacts with potentially globally catastrophic consequences.

Stellar phenomena such as supernova explosions or gamma ray bursts could also render the Earth's environment less habitable or uninhabitable through major depletion of the ozone layer; such events happen in the Earth's vicinity once every 250–500 million years (Melott and Thomas, 2011). Increases in solar luminosity will render the Earth uninhabitable to animal species within several billion years. While this would not seem survivable for present human civilisation, on such timescales, humans may have developed adaptations or spread beyond the Earth.

Using the track record of survival for the genus *Homo*, Snyder-Beattie et al. (2019) estimate annual probability of extinction from exogenous risks of below 1 in 870,000 per year. However, Baum (2023) cautions that this estimate does not take into account the interaction between threats *Homo* historically faced and the various systemic fragilities of modern human civilisation.

## Endogenous threats: Caused or closely linked to human activity

### Climate change and environmental and ecological degradation

Anthropogenic activity is leading to an unprecedented acceleration of global temperature. Scenarios involving greater temperature rise are predicted to result in increasingly severe droughts, flooding, heatwaves, famines, disease spread and loss of biodiversity, with major global human mortality implications (Lee et al., 2023). The likelihood of catastrophic outcomes depends in part on the rate at which global $CO_2$ emission reduction targets are met and on the Earth's climate sensitivity. The sixth assessment report of the Intergovernmental Panel on Climate Change (IPCC) narrows the "likely" range of climate sensitivity to 2.5–4 °C, meaning an expected temperature rise in this range for a doubling of $CO_2$ levels compared to pre-industrial levels (Lee et al., 2023). This suggested a reduction of the likelihood of the global catastrophic scenarios that would be associated with greater temperature rises. However, there remains substantial uncertainty around many aspects of climate sensitivity, including the impact of positive feedback loops such as possible methane release from thawing permafrost and the

---

[1]For further discussion on existential risk definitions, see Torres (2023a).

[2]Definitions typically vary by threshold of severity; for example, Bostrom and Cirkovic (2011) propose that a disaster costing 10 million lives or 10 trillion dollars of damage would be a global catastrophe; the Global Challenges Foundation uses a threshold of 10% of the global population (https://globalchallenges.org/global-risks/).

[3]For a history of extinction and existential risk studies, see Beard and Torres (2024); for a recent bibliometric analysis of global catastrophic risk research, see Jehn et al. (2024).

---

[4]Ord (2020) concludes that it is possible humanity would survive a 10-km asteroid impact but that extinction risk is high; for 1- to 10-km asteroids, extinction risk is lower but cannot be ruled out.

potential for tipping points to be passed resulting in a self-perpetuating change in the Earth's system (Armstrong McKay et al., 2022).

Some extreme possibilities associated with high temperature and $CO_2$ emission scenarios have yet to be ruled out; for example, a recent model predicted stratocumulus cloud break-up in 1300 ppm scenarios leading to an additional 8 °C rise, although the authors note substantial uncertainties remain (Schneider et al., 2019). The potential for climate change to bring about outcomes as severe as human extinction or civilisational collapse remains the subject of academic debate. Ord (2020) and MacAskill (2022)[5] conclude that the direct risk of human extinction from climate change appears very low. Kemp et al. (2022) argue that potential worst-case outcomes for climate change, including global catastrophe or human extinction, remain "dangerously unexplored," highlighting that in addition to direct impacts, climate change may exacerbate other global threats.

Climate change is exacerbating ongoing anthropogenic biodiversity loss and environmental degradation. Other drivers of biodiversity loss include habitat destruction and fragmentation, pollution and overexploitation of species and resources (IPBES, 2019). Dasgupta (2021) reviews the extensive dependence of human society on ecosystem services; substantial global harm is to be expected from the degradation of these services, although a direct path to human extinction is not outlined. Surveying the literature Kareiva and Carranza (2018) note a "growing scepticism regarding the strength of evidence linking trends in biodiversity loss to an existential risk for humans," arguing that in the absence of unlikely circumstances such as 90% of biodiversity being lost, human civilisation would survive. Ord (2020) argues that ongoing environmental damage is a source of unknown threats to humanity and that more research is needed.

Under scenarios of unsustainable anthropogenic activity, wide-scale pollution, resource depletion, soil erosion, deforestation and other consequences of human activity may make global civilisation considerably less resilient, but a direct link to full extinction appears implausible. Concerns have been raised about the widespread release of chemicals that may be harmful to human health; in particular endocrine disruptors, which have been linked to fertility decline (Gonsioroski et al., 2020; Swan and Colino, 2022). While it is premature to consider these extinction threats, more research is needed on their impacts on populations and reproductive health. Conversely, concerns have been raised for half a century that rising global population might lead to civilisational collapse (Ehrlich and Ehrlich, 2009); however, research increasingly suggests that global population will peak and begin to decline within this century (Adam, 2021).

### Biological agents

Infectious disease has been a contributing factor in 3.7% of species extinctions known to have occurred between the years 1,500 and 2008 (Smith et al., 2009). In humans, the 1918 Spanish influenza outbreak killed 1–5% of world population (Johnson and Mueller, 2002, Spreeuwenberger et al. 2018).[6] Koch et al. (2019) estimate that epidemics killed 90% of the indigenous population of the Americas as the result of diseases carried over by European colonists in the fifteenth and sixteenth centuries.

It is considered very unlikely that a naturally occurring pandemic[7] would be a primary cause of human extinction (Ord, 2020), and the author is aware of no credible work presenting plausible pathways. Such a disease would need both to spread to nearly all humans across the world and be of extremely high lethality or cause infertility at extremely high rates. However, it seems plausible that pandemic disease could play a contributing role in combination with other factors in an extinction scenario.

A common concern is that advances in the biological sciences and biotechnology could enable the creation of more dangerous viruses and other forms of biological agent than exist in nature (Millett and Snyder-Beattie, 2017). Biological weapon development and use are prohibited under the Biological Weapons Convention; however, leading states have developed major biological weapons programmes in the past, and it is conceivable that such directions might be pursued in future or are presently being pursued clandestinely (Riedel, 2004). Further, advances in these sciences lower the barrier to entry to the development of dangerous agents in terms of cost, expertise and equipment needed, potentially bringing globally catastrophic capabilities within the reach of small groups in future. While groups willing to bring about global catastrophe or even extinction would be expected to be rare, there have been notable examples of groups motivated by near-apocalyptic ideology such as the Aum Shinrikyo cult,[8] which attempted to weaponise Anthrax and carried out successful local nerve agent attacks (Lifton, 1999; Torres, 2016). Gopal et al. (2023) outline two scenarios leading to civilisational collapse: a "wildfire" scenario in which one or more pandemic agents leads to disruption of food, water, power supply and law enforcement and a "stealth" scenario in which an initially asymptomatic agent with long incubation time is spread to a majority of humanity. They consider both scenarios most likely to involve accidental or deliberate release of human-modified agents. Direct targeting of humans is only one plausible pathway to global harm; Monica Schoch-Spana et al. (2017) note the potential pathogens targeting widespread eradication of food sources; and other forms of environmental harm are plausible.

The capability to re-engineer biology more fundamentally may create unprecedented risks. In 2024, a group of leading biological scientists called for a halt to research to develop "mirror bacteria," designed to have an opposite chirality to that of all forms of existing life (Adamala et al., 2024). The group argues that it is likely these synthetic life forms would both resist predation and bypass immune systems in humans, animals and plants, potentially leading to risk "of unprecedented scope and scale" due to invasive spread and widespread lethal infection.

While robust likelihoods are not possible here, Millett and Snyder-Beattie (2017) deem the likelihood of full human extinction primarily from a bioweapon or engineered biological agent as "extremely low" but not ruled out, although Ord (2020) estimates as high as 1 in 30 for the coming century. Again for extinction, scenarios involving one or multiple pathogens in combination with other factors may be most plausible.

---

[5]Drawing on research by Halstead (2022).
[6]Global Mortality estimates range from 17.4 million to 100 million.

[7]Natural pandemics are included within endogenous threats, as their spread is substantially mediated by human-environment and human-human interaction.

[8]Aum Shinrikyo's ambition involved wiping out all of humanity except the several thousand members of the cult itself.

## Nuclear war

The most credible global catastrophe scenarios resulting from nuclear war involve large-scale nuclear weapon exchange bringing about a catastrophic climate effect known as a nuclear winter. In these scenarios, smoke from burning cities, industrial facilities and oilfields enters the upper atmosphere, disrupting sunlight for a period of years and leading to wide-scale crop failures and famine. Disagreement exists between scientific teams modelling the climatic impact of nuclear exchanges. Xia et al. (2022) estimate that over 2 billion people could die from nuclear war between Pakistan and India and over 5 billion between Russia and the USA. However, Reisner et al. (2019) predict more limited generation and dispersal of soot, leading to less severe outcomes.

The size of nuclear warhead arsenals has been reduced from a combined global peak of over 70,000 in the 1980s (Kristensen and Norris, 2013), but ~12,512 still exist, with the USA holding 5,244 and Russia holding 5,889 (Kristensen and Korda, 2023). Recent years have seen increasing geopolitical tensions between nuclear powers, heightening of political rhetoric around nuclear weapons use, the non-renewal of key treaties such as New START (Williams, 2023) and the development of new technology that might upset strategic balance (Bajema and Gower, 2022), leading to concerns that the risk of nuclear war remains high. A nuclear war appears unlikely to bring about extinction directly, but would severely undermine global ability to respond to other threats.

## Artificial intelligence

Three plausible pathways to global catastrophe mediated by artificial intelligence (AI) have been proposed (Hendrycks et al., 2023). The first relates to AI being used by humans to enact more powerful offensive warfare than in previous history or contributing to military escalation scenarios (Maas et al., 2023). The second relates to AI accelerating development of other sciences and technologies such as bioscience (Sandbrink, 2023) relative to human understanding and governance, increasing any risks posed by these developments. The most discussed scenario, and that most directly linked to human extinction, involves the development and subsequent loss of control of artificial general intelligence (AGI) with autonomy and superhuman cognitive and strategic planning ability. The creation of AGI that is fully aligned with human goals and values is considered by many AI scientists a key challenge for future AI development (Russell, 2019); a highly capable system that was not fully aligned might, for example, perceive human oversight as an obstacle to be bypassed and pursue actions (for example, large-scale modification of the Earth's environment) that might lead to human extinction as a result of indifference to human survival. A rapidly growing number of research leaders consider the development of AGI feasible in coming decades or even years, and an open letter signed by many leading technologists in 2023 stated *"Mitigating the risk of extinction from AI should be a global priority alongside other societal-scale risks such as pandemics and nuclear war."*[9] The topic remains controversial, with many research leaders considering such outcomes highly unlikely.[10] A recent survey of 2,778 AI authors found that between 37.8% and 51.4% (depending on framing of the question) of respondents gave at least a 10% probability to advanced AI causing human extinction or an equivalently bad outcome (Grace et al., 2024).

## Physics experiments and unknown future technological developments

Concerns have been raised that particle physics experiments might have the potential to create consequences that could destroy the Earth: for example, creating a black hole, strangelets[11] or triggering a phase transition that would rip the fabric of space (Rees, 2021). According to Rees, the most favoured physics theories imply that the risks from such experiments "within our current powers" are zero; however, alternative theories imply the possibility of these catastrophic outcomes. Collisions between cosmic ray particles of much higher energies occur frequently without catastrophic consequence, offering some evidence against such concerns (Hut and Rees, 1983). Kent (2004) discusses and critiques risk analysis estimates for the RHIC supercollider, for which probabilities of catastrophe ranged from $10^{-5}$ to $2 \times 10^{-8}$ (Dar et al., 1999; Jaffe et al., 2000).

Concerns have also been raised about theorised but as yet undeveloped technologies such as atomically precise manufacturing (Phoenix and Drexler, 2004). More generally, many scholars have noted that some of the most powerful and potentially destructive technologies have been developed relatively recently in human history (Rees, 2003; Bostrom, 2013; Ord, 2020). Given expectations of continued scientific advancement, it is likely that as-yet undeveloped technologies will pose catastrophic risks in coming decades and centuries and plausibly represent the greatest direct risk of human extinction.

## Multiplicative stresses and civilisational vulnerability

A common theme across many of the risks discussed is that they may cause global catastrophe but are unlikely to wipe out the human species in isolation. However, it is realistic to expect that some catastrophes may trigger further catastrophic consequences across the categories above or otherwise occur in combination. The severity of some past major mass extinctions likely resulted from reinforcing interactions between multiple stresses, some playing out over thousands of years. The Permo-Triassic saw a cascade of global warming, ocean acidification and anoxia, and methane and hydrogen release Bond and Grasby (2017). The Cretaceous-Paleogene saw the catastrophic consequences of the Chixhulub impact devastating a global environment already stressed by the Deccan Traps (Keller, 2012).

Returning to modern times and shorter timescales, Kareiva and Carranza (2018) argue that "it is the interconnections of stresses and the way we respond to environmental shocks that promulgates the greatest existential risk.". Rees (2003) draws attention to the fragile nature of global supply chains: a catastrophe that resulted in higher order impacts, such as disruption of power and food supply chains, would quickly render cities uninhabitable. A civilisation reeling from one catastrophe might also be less resilient to further threats: a planet reeling from nuclear war may not be able to muster

---

[9]https://www.safe.ai/statement-on-ai-risk accessed 29 December 2023.

[10]For example, see the Science Media Centre's expert reactions to the AI extinction risk statement https://www.sciencemediacentre.org/expert-reaction-to-a-statement-on-the-existential-threat-of-ai-published-on-the-centre-for-ai-safety-website/ (accessed 3 January 2024).

[11]A strangelet is a hypothetical strange matter particle; it is speculated that strangelets could be extremely stable and convert normal matter to strange matter on contact, which could have catastrophic consequences if created in a heavy ion collider (Jaffe et al., 2000).

the resources to embark on a project to divert an asteroid. Nonetheless, total human extinction remains a very high bar, even for multiple of these threats in combination.

Analysing overall civilisational vulnerability and higher-order impacts of global-scale catastrophes is challenging. There is growing work in this direction. Baum and Handoh (2014) propose a "Boundary Risk for Humanity and Nature" framework that combines concepts from global catastrophic risk and the planetary boundaries framework. This intersection is further explored in studies addressing the relationship between the sustainable development goals and existential risk (Cernev and Fenner, 2020) and the role of planetary boundary breach in global catastrophe scenarios (Cernev, 2022). Jehn (2023) examines the role of planetary boundary breach in exacerbating or mitigating the cascading impacts of nuclear war.

Liu et al. (2018) present a taxonomy of existential hazards grouped by how global vulnerability and exposure to the hazards plays out. Avin et al. (2018) analyse severe global catastrophic risk mechanisms by critical systems affected, global spread mechanism and prevention and mitigation failure. Cotton-Barratt et al. (2020) propose a three defence layer model against extinction-level catastrophes, focusing on prevention, response and resilience. Kemp et al. (2022) outline a research agenda around climate-triggered catastrophes and societal fragility. Maher and Baum (2013), MacAskill (2022) and Belfield (2023) explore factors that might contribute to either extinction or recovery following a civilisational collapse. Drawing on a range of concepts in systemic risk, Arnscheidt et al. (2024) argue that emergent properties of the global system make important contributions to the risk of global catastrophic outcomes, noting that a focus on hazards in early global catastrophic risk research critically neglects factors such as amplification and vulnerability. Further valuable insights are likely to come from collaboration between existential risk, systemic risk and collapse research.[12]

## Estimating the unprecedented

I have used the term "estimate" in this review for expert assessments of extinction risks posed by hazards ranging from novel bioweapons through to physics experiments. These assessments are provided by scholars with substantial expertise, but there is frequently limited empirical evidence to draw on. Research on human extinction invariably involves at least one of three methodological challenges that make the assignment of robust probabilities difficult (for an analysis of likelihood estimation methods for existential hazards, see Beard et al., 2020). A first category of risks involves extremely rare events, where there is limited evidence to go on. A second category relates to entirely unprecedented developments, for which we must often primarily draw on experts' subjective judgements. A third challenge relates to scenarios involving multiple interacting risks or trends; the complexity of these scenarios makes them challenging to assign likelihood to. Plainly put, humans are quite different from other species. We have no similar extinctions to draw on and so must fumble in the dark with what we have.

In *The Precipice*, Toby Ord judges the overall likelihood of an existential catastrophe in the coming century at about one in six. This figure is itself informed by subjective judgements across a range of individual risks where certainty is not possible. Ord caveats the purpose of these judgements as showing "the right order of magnitude, rather than a more precise probability"; even so, it would be reasonable to regard them as incorporating some element of guesswork given the presently unknowable nature of many of the informing variables. The majority of Ord's overall estimate is contributed by Ord's estimate of a one in ten likelihood of extinction from unaligned AI, with engineered pandemics and other anthropogenic risks making up most of the remainder. Even with these latter categories, Ord's estimates have been critiqued as being too high for climate- and engineered pandemic-caused extinction (Thorstad, 2023a, 2023b). However, Ord's overall estimate is similar to that derived from a 2008 survey of global catastrophic risk experts, which provided an overall risk of extinction of 19% prior to 2,100 (Sandberg and Bostrom, 2008). The majority of this survey's overall risk estimate was also linked to emerging or future technological development. These are also subjective estimates, and the previous discussion applies.

## Ethical and social dimensions of human extinction

The social, ethical and political implications of human extinction and its study are hotly debated. Some scholarship emphasises the importance of reducing human extinction risk drawing on frameworks such as longtermism, which places a high moral value on the lives of unborn future generations and future achievements of humanity (MacAskill, 2022). Critics contend that a focus on extinction of humanity as a whole or on the value of hypothetical future lives ignores or may detract attention from ongoing catastrophic impacts on existing peoples, particularly marginalised and indigenous peoples (Torres, 2023a,b; Mitchell and Chaudhury, 2020).

Bostrom (2013)'s commonly used definition of existential risk includes both extinction and destruction of Earth's desirable future potential as outcomes to avoid. "Desirable future potential" is open to interpretation within the definition; however, Bostrom's own articulations of desirable futures have been criticised as heavily influenced by transhumanism and unlikely to be shared as an ideal by a majority of current people worldwide (Cremer and Kemp, 2024). It has also been argued that human extinction is a dangerous motivator: such high stakes could be used to justify extreme measures such as pre-emptive military action or authoritarian governmental action (Bostrom, 2019; Cremer and Kemp, 2024).

Many themes overlap with discussions of the Anthropocene,[13] and similar critiques have been raised in relation to both literatures. It is argued that a universalist narrative portraying humanity as a single, unified agent risks obfuscating inequalities in responsibility for and vulnerability to present crises such as climate change (Todd, 2015; Head, 2016), as well as responsibility for harms such as the legacies of colonialism (Whyte, 2017; Yusoff, 2018). Extractive practices associated with Global North countries have made disproportionate contributions to a range of environmental problems; it has similarly been argued that risks posed by nuclear weapons arsenals (Biswas, 2014) and advanced AI (Abungu et al., 2024) are borne by countries worldwide, while responsibility for their production lies mainly with a subset of Global North nations. Debates over the role of capitalism feature in both literatures. Moore (2015) proposes the Capitalocene as an alternative framework to the

---

[12]See *How Worlds Collapse* (eds. Centeno et al., 2023) for a recent edited volume combining scholarship across these fields.

[13]The validity of the Anthropocene concept has been heavily contested and was recently rejected by the International Union of Geological Scientist's Anthropocene Working Group (Witze, 2024).

Anthropocene, arguing that the ecological crisis stems from specific patterns associated with capitalism. Ware (2024) similarly centres capitalism and the pursuit of economic growth as the driver of a range of crises threatening humanity. In contrast, Aschenbrenner and Trammell (2020) argue that over a long time frame, continued growth is likely to contribute to a richer global society better able to allocate resources to its safety.

A detailed treatment of these complex debates is out of scope for this review. However, they are important considerations, especially as calls grow for extinction risk mitigation to be incorporated into global policy decisions (Boyd and Wilson, 2020).

## Conclusion

The literature suggests that the likelihood of human extinction from exogenous threats, where we have more robust data to draw on, is extremely low. For anthropogenic threats, there is far greater uncertainty. For many anthropogenic threats such as nuclear war, pandemics and environmental degradation, the likelihood of such events and the scale of their catastrophic consequences strongly justify prevention and mitigation efforts. However, it is unclear how plausible they are as causes of extinction of *homo sapiens*. Through the lens of civilisational resilience, however, it can be argued that a global civilisation that avoids or robustly addresses these catastrophic threats will be far better placed to respond to any extinction-level threats it encounters; in terms of available resources, ability to coordinate and foresight and governance tools to identify and address emerging threats.

This argument carries particular weight if, as many experts believe, the greatest risks are anthropogenic and lie ahead of us; stemming from a global civilisation with a greater capacity to alter or degrade its environment deliberately or unintentionally; and rapidly developing technologies with novel characteristics that are more powerful than those we have experience with. If so, having the foresight, governance and response capacities necessary at a scientific, political and civil society level will be crucial. Humans may be hard to wipe out, but that will not stop us giving it our best shot.

**Open peer review.** To view the open peer review materials for this article, please visit http://doi.org/10.1017/ext.2025.4.

**Acknowledgements.** I am grateful to Constantin Arnscheidt, Emma Bates, John Alroy, Barry Brook and three anonymous reviewers for helpful comments that greatly improved this article.

**Competing interest.** The author declares none.

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
