## [Reviewer Report]

This is a competent, albeit quite brief, overview of the current literature on the potential causes of human extinction. The author is clearly familiar with the relevant literture, and does a nice job of summarizing key ideas within that literature. My only critique is that I am not sure exactly what this adds to the field: there are already a number of synoptic literature reviews out there, some of which are quite recent (e.g., “The Era of Global Risk,” by Rees et al.). Despite this hesitation, I am opting for “accept” with no revisions.

---

## [Reviewer Report]

This is an excellent review article: lucid, knowledgeable, and judicious. The author takes an admirably balanced approach to this complex and controversial topic, offering a valuable synthesis of the field without making any tendentious claims. The essay is well structured and covers the literature on both exogenous and endogenous threats thoroughly and economically. I appreciated the author’s acknowledgement of the limitations of the discussion, including the potential for moral hazard in focusing on future extinction rather than present-day suffering (4). I sometimes find that work in this field (e.g. Ord 2020) unhelpfully describes the human species as if it was a single unified agent, but generally this article avoids that problem.

I certainly recommend the article for publication, but have a few minor queries.

I think it’s generous to describe the guesses made about the likelihood of human extinction as ‘estimates’. Given the numerous factors and uncertainties involved, I find it hard to take Ord’s ‘one in six’ seriously. Perhaps I’m quibbling, but at what point does a ‘subjective estimate’ (10.9) become a guess?

While I could not see any obvious gaps in the works cited, the author might in future want to look into recent work on human extinction as an idea e.g. E Torres 2024, Ware 2024.

1.16 There is a hyphen before ‘unlikely’. I think a dash is required.

1.17 What evidence is there that ‘decision-makers’ are likely to factor in concerns about human extinction as compared to more directly pressing issues?

3.25-30 The word ‘advances’ is used three times in quick succession. This is a little inelegant; also, the term ‘developments’ might be preferable as less value-laden in the context of science.

3.40 ‘Desirable’ for whom?

6-7 I think a bit of rephrasing is needed in the ‘Climate change, environmental and ecological degradation’ section. In particular, it needs to be clear that biodiversity loss is not only a consequence of climate change but also caused and accelerated by separate factors, such as changing land use, pollution, etc. The importance of defaunation might also be acknowledged (see e.g. Dirzo et al. 2014).

8.14 ‘Strangelets’ may need glossing (the word was new to me).

9.32 I was unaware of the Baum & Handoh 2014 article combining ‘concepts from global catastrophic risk and the planetary boundaries framework’. Given the increasingly profile of the latter, I wondered if anyone else had developed this potentially valuable combined approach.

12 Missing line space between the references to Dasgupta and Ehrlich.

Throughout, superscript numbers indicating footnotes are placed before punctuation marks when they should be placed after.

---

## [Reviewer Report]

Thank you for the opportunity to review this manuscript - it is ambitious in aim and scope, which is great to see. There are areas to work on in a revision that will enable this manuscript to better achieve its goals.

First, the conceptual work on the Anthropocene is suggested but not made explicit in the manuscript. It is included as a keyword and is clearly implied in the topic of the paper, but the vast body of research on extinction in the Anthropocene is barely engaged with and the import of this concept (that has recently been voted down by the scientists trying to evaluate its value as a label for new epoch).

Second, while the Anthropocene is not rigorously engaged with as a scientific concept or term that is still being debated, the critical literature trying to tease out the implications of this universalist thinking is also not. For example, this reference to how Indigenous people experienced massive loss of life as a result of colonisation is a demonstration of a lack of critical engagement with the social scientific literature on the Anthropocene: ‘Koch et al. (2019) estimate that epidemics killed 90% of the indigenous population of the Americas over the century following the European arrival in 1492.’ The epidemics arrived with colonisers: they were a result of settler colonial practices. If Zoe Todd’s or Kyle Powys Whyte’s insightful contributions to critiques of the Anthropocene had been grappled with properly here, then the absolving of responsibility and the glossing over the impacts of colonisation would not have happened.

Third, the opening premise setting the scene in this manuscript draws on a claim about how modern human activity has affected species across the world: ‘Modern human activity is estimated to have accelerated species extinction rates across the Earth’s biosphere to between 100 and 1,000 times higher than their background rate over the past tens of millions of years (Dasgupta, 2021)’. To present the current biodiversity extinction wave as a result of all modern human activity absolves responsibility for the uneven extractive practices of the Global North and the inequitable distribution of the burdens of environmental harms in one universalist claim. Again, if a critical engagement with scholarship on what the Anthropocene is (or is not) had grounded this paper, such a claim would not be possible. For instance, Lesley Head and Chris Gibson have written extensively on the impacts of ‘the moderns’ and offer critical readings of universalising Anthropocene thinking - unpacking it as a contingent lived reality. Further, Jessica McLean does this work too.

The paper is well written and engaging but framed in such a way that it misses major key arguments and overlooks vast literatures on the very thing it is trying to examine. I hope the author/s take the opportunity to engage with this work in a revision as these are important topics that we need to read more about.

---

## [Editor Report]

The manuscript “Extinction of the Human Species: What Could Cause It and How Likely Is It to Occur?” presents a timely overview of the literature on potential pathways to human extinction, covering both exogenous and anthropogenic threats. The author synthesises a broad spectrum of research, providing valuable insights into the likelihood of various extinction scenarios and acknowledging the limitations inherent in such speculative analyses. For a reader new to the topic, this paper presents an excellent knowledge grounding and primer.

Reviewer 1 recognises the author’s familiarity with the relevant literature and the concise summary provided but questions the manuscript’s novelty. Reviewer 2 commends the balanced approach and thorough coverage but suggests clarifying the speculative nature of extinction likelihood estimates and advises minor stylistic improvements and updates to the literature. Reviewer 3 urges a more explicit engagement with the Anthropocene and its social and ethical dimensions, recommending a critical examination of universalist claims and the inclusion of diverse perspectives on environmental responsibility.

In light of these reviews, I recommend a major revision to address the following core requirements:

1. Differentiate this review from existing literature by clearly articulating its unique contributions to the field.

2. Clarify the speculative nature of likelihood estimates for human extinction and consider the semantic distinction between ‘estimates’ and ‘guesses’.

3. Engage more explicitly with the concept of the Anthropocene, incorporating critical perspectives on its implications and the role of human activity in biodiversity loss.

4. Expand the discussion on the social and ethical dimensions of human extinction, integrating insights from indigenous studies and critical social science to challenge universalist narratives.

5. Make minor stylistic and formatting corrections as suggested by Reviewer 2, including the use of dashes, repetition of terms, and footnote placement.

These revisions will strengthen the manuscript’s contribution to the field, providing a more nuanced and comprehensive review that acknowledges the complexity and ethical considerations surrounding the topic of human extinction.

Barry Brook (acting as Handling Editor)

---

## [Editor Report]

Following the thorough revisions undertaken by the author, I have re-examined the manuscript entitled “Extinction of the Human Species: What Could Cause It and How Likely Is It to Occur?” and cross-checked the response against the reviewer feedback. 

In my view as Handling Editor, the updated version nicely addresses all substantive critiques, including expanding discussions on multiplicative stresses and tipping points, refining the scope and relevance of cited literature (such as clarifications around the ‘Big Five’ mass extinctions), and adding a dedicated section on ethical, social, and political dimensions of human extinction. The author has also satisfactorily responded to smaller editorial concerns, such as minor stylistic points and clarifications of assumptions around extinction likelihood estimates.

Given these updates, I am confident the manuscript now meets the journal’s standards and provides a clear, incisive, and balanced assessment of critical issues around human extinction. It should stand as an important, accessible summary of the human-extinction literature for the general reader. 

Each reviewer’s central concerns have been carefully addressed, and no further technical or conceptual revisions appear necessary. I therefore recommend the manuscript be accepted for publication without requiring additional rounds of review.